# Management and Prognosis of Cardiac Metastatic Merkel Cell Carcinoma: A Case–Control Study and Literature Review

**DOI:** 10.3390/cancers14235914

**Published:** 2022-11-30

**Authors:** Tomoko Akaike, Kelsey Cahill, Gensuke Akaike, Emily T. Huynh, Daniel S. Hippe, Michi M. Shinohara, Jay Liao, Smith Apisarnthanarax, Upendra Parvathaneni, Evan Hall, Shailender Bhatia, Richard K. Cheng, Paul Nghiem, Yolanda D. Tseng

**Affiliations:** 1Division of Dermatology, Department of Medicine, University of Washington, Seattle, WA 98195, USA; 2Department of Radiology, University of Washington, Seattle, WA 98195, USA; 3TRA Medical Imaging, Tacoma, WA 98402, USA; 4Clinical Research Division, Fred Hutchinson Cancer Center, Seattle, WA 98109, USA; 5Department of Radiation Oncology, University of Washington, Seattle, WA 91895, USA; 6Division of Medical Oncology, Department of Medicine, University of Washington, Seattle, WA 98195, USA; 7Division of Cardiology, University of Washington Medical Center, Seattle, WA 98195, USA

**Keywords:** Merkel cell carcinoma, cardiac metastasis, immunotherapy, radiotherapy, case–control study, literature review, prognosis, cardio-oncology

## Abstract

**Simple Summary:**

Approximately 20% of patients with Merkel cell carcinoma (MCC) will develop distant metastasis. Rarely, MCC metastases may involve the heart; there are limited data on management and prognosis of cardiac metastasis of MCC. Among a prospective registry of 582 patients with metastatic MCC (mMCC), we identified 9 patients (1.5%) with cardiac involvement. We found that cardiac mMCC most commonly involves the right heart (8 of 9; 89%) and occurs relatively late in the disease process (median 925 days from the initial diagnosis to cardiac involvement). In our cohort, cardiac mMCC frequently responds to immunotherapy and cardiac radiotherapy, which can both be delivered with minimal cardiac toxicity. Cardiac involvement was not associated with worse survival compared to MCC patients with non-cardiac distant disease. These results are timely as cardiac mMCC may be increasingly encountered in the era of immunotherapy as patients with metastatic MCC live longer.

**Abstract:**

Merkel cell carcinoma (MCC), an aggressive neuroendocrine skin cancer, has a high rate (20%) of distant metastasis. Within a prospective registry of 582 patients with metastatic MCC (mMCC) diagnosed between 2003–2021, we identified 9 (1.5%) patients who developed cardiac metastatic MCC (mMCC). We compared overall survival (OS) between patients with cardiac and non-cardiac metastases in a matched case–control study. Cardiac metastasis was a late event (median 925 days from initial MCC diagnosis). The right heart was predominantly involved (8 of 9; 89%). Among 7 patients treated with immunotherapy, 6 achieved a complete or partial response of the cardiac lesion. Among these 6 responders, 5 received concurrent cardiac radiotherapy (median 20 Gray) with immunotherapy; 4 of 5 did not have local disease progression or recurrence in the treated cardiac lesion. One-year OS was 44%, which was not significantly different from non-cardiac mMCC patients (45%, *p* = 0.96). Though it occurs relatively late in the disease course, cardiac mMCC responded to immunotherapy and/or radiotherapy and was not associated with worse prognosis compared to mMCC at other anatomic sites. These results are timely as cardiac mMCC may be increasingly encountered in the era of immunotherapy as patients with metastatic MCC live longer.

## 1. Introduction

Merkel cell carcinoma (MCC) is a rare neuroendocrine skin cancer. Risk factors for MCC include advanced age, fair skin, and chronic immunosuppression such as chronic lymphocytic leukemia, human immunodeficiency virus, or prolonged immunosuppressive agents for autoimmune disease or maintaining organ transplantation [1,2,3,4,5,6]. In the United States, most MCC cases (80%) are caused by the Merkel cell polyomavirus (MCPyV), with a minority (20%) associated with ultraviolet-induced point mutations from extensive sun exposure [7,8]. About 40% of MCC patients develop recurrent disease, with 20% developing distant metastasis, typically within the first 2 years after initial treatment [9]. The recurrence rate following definitive therapy is notably higher than that of invasive melanoma (19%), cutaneous squamous cell carcinoma (5–9%), or basal cell carcinoma (1–2%) [10,11,12]. MCC spread is often sequential, initially affecting the draining regional lymph nodes and/or in-transit lymphatics prior to involvement of distant sites. Common sites of initial distant MCC metastases are non-regional lymph nodes (41%), followed by skin/body wall (25%), liver (23%), and bone (21%) [13].

MCC metastasis to the heart is rare, with experience limited to isolated case reports [14,15,16,17,18,19,20,21,22,23,24,25,26,27]. Before the era of immunotherapy, the prognosis of metastatic MCC (mMCC) was limited to months [9]. However, with improved imaging and availability of immunotherapy leading to longer survival, this uncommon metastasis may be increasingly encountered.

Given the anatomic location, cardiac mMCC poses diagnostic and therapeutic challenges, and optimal management of cardiac mMCC is unclear. Within a large prospective registry, we reviewed patient and imaging characteristics, management, and outcomes of MCC patients who developed metastasis to the heart. We also conducted a review of previously published cases of cardiac mMCC. As a secondary objective, we explored whether cardiac mMCC is associated with a worse prognosis by matching our cases to a control cohort. To the best of our knowledge, this represents the largest reported cohort of mMCC to the heart and its management.

## 2. Materials and Methods

### 2.1. Patient Cohort and Eligibility

We queried a prospective MCC observational registry, which included patients with pathologically confirmed MCC. These patients were enrolled after informed consent between February 2003 and October 2021. The date of data lock was 29 October 2021. The registry was approved by the Institutional Review Board (IRB) at Fred Hutchinson Cancer Research Center (FHCRC IRB #6585, Seattle, WA, USA). Established protocols were followed for data entry and updates. At least annually, patients were contacted by email and/or phone for changes in disease status, recurrence/progression, and treatments. Patients with missing treatment details or inadequate follow-up to assess response and/or survival were excluded from the cohort. A matched reference cohort of patients with non-cardiac metastases was also selected from the registry, as described in the statistical analysis section. Patients were staged following the guidelines of the American Joint Cancer Committee (AJCC) eighth staging system [28].

### 2.2. Analysis of Pathological Data

Pathology of the primary site for all patients was independently reviewed by pathologists at our institution to confirm an initial diagnosis of MCC, with the exception of one patient. For the single patient whose specimens were not available for institutional pathological specimen review, the pathological report was reviewed to verify MCC diagnosis. MCPyV status was determined either by MCPyV oncoprotein antibody serology assay [29] or immunohistochemistry using anti-MCPyV T-antigen antibody (CM2B4) [30]. Pathological data were collected in cases with biopsy-confirmed cardiac mMCC.

### 2.3. Analysis of Radiological Imaging Studies

A single radiologist with expertise in both cardiothoracic imaging and nuclear medicine retrospectively reviewed pertinent imaging to confirm the location of cardiac metastasis. In addition, imaging studies at the time of and immediately prior to cardiac metastasis were reviewed to identify concurrent sites of metastasis. Radiological assessment of response was determined by Response Evaluation Criteria in Solid Tumors (RECIST) version 1.1 [31].

### 2.4. Literature Search

We performed a literature search using PubMed (National Center for Biotechnology Information, Bethesda, MD, USA), conducted on 19 May 2022. Search keywords included “Merkel cell carcinoma”, “cardiac”, “heart”, “metastasis”, or “metastatic”. Non-English reports were excluded, except 1 French report with an English abstract. Additionally, excluded were reports of cases that metastasized only to pleura or pericardiac lymph nodes. One case report was also excluded because the same patient had previously been reported.

### 2.5. Statistical Analysis

Descriptive analyses were performed to summarize clinical and patient data. Overall survival (OS) time was defined as time of cardiac mMCC diagnosis to all-cause mortality; OS rates and median OS was estimated using the Kaplan–Meier estimator. Duration of local control was defined as time from cardiac mMCC diagnosis to time of progression at the site of the initial treated cardiac lesion with death considered as a competing risk. Local control rates and median duration of local control was estimated using the empirical cumulative incidence function [32]. Outcomes were censored at the time of the last follow-up available by the study cut-off date.

To explore whether patients with cardiac mMCC have worse OS compared to patients with non-cardiac mMCC, we selected a reference cohort with non-cardiac mMCC from the same prospective MCC observational registry. The reference cohort was matched to the cardiac mMCC patients using the following covariates that may influence OS: immune suppression status, age (±10 years), sex, disease status of stage IV at initial diagnosis (versus at relapse/progression), and number of prior metastatic episodes. OS time for the non-cardiac mMCC patients was defined as the time from the matching non-cardiac mMCC diagnosis to all-cause mortality to be comparable with the definition of OS used for the cardiac mMCC patients. The matching non-cardiac mMCC diagnosis was the one with the same number of prior metastatic episodes as the matched cardiac mMCC patient at the time of the cardiac mMCC diagnosis. Because of the small sample size of cardiac mMCC patients, all available non-cardiac mMCC matches were included for each cardiac mMCC patient. Calculations were weighted to account for a variable number of matches per cardiac mMCC patient [33]. A detailed description of the selection of the reference cohort is provided in the Appendix A. The non-parametric bootstrap was used to calculate 95% confidence intervals (CIs) for the cardiac and non-cardiac mMCC OS curves [34]. The OS curves were compared using the stratified log-rank test, and thus, each cardiac mMCC patient was only compared with matching non-cardiac mMCC patients [35,36] All statistical calculations were conducted with the statistical computing language R (version 4.0.3; R Foundation for Statistical Computing, Vienna, Austria).

## 3. Results

### 3.1. Patients and Tumor Baseline Characteristics

Among 1625 patients with MCC in our registry, 582 patients (36%) had distant metastases identified either at the time of initial MCC diagnosis (*n* = 130; 8%) or after initial treatment on surveillance (*n* = 452; 28%). Among these 582 patients, 9 (1.5%) had mMCC to the heart. One cardiac metastasis was identified at initial diagnosis, while the other 8 developed after initial diagnosis (range 302–1494 days). Patient baseline characteristics at the initial MCC diagnosis are summarized in Table 1. The median age at initial diagnosis was 69 (range 52–86), with a male predominance. The most common initial primary site was the extremity (*n* = 4; 44%). Most patients had local/regional disease at diagnosis (1 stage I, 6 stage III), and the majority were MCPyV-positive.

### 3.2. Cardiac Metastatic MCC

In all 9 patients, cardiac mMCC was detected incidentally on surveillance or baseline imaging. Cardiac mMCC was first noted on fluorodeoxyglucose (FDG) positron emission tomography/computed tomography (PET/CT) with intensive FDG uptake in 6 patients and on contrast CT in 3 patients (Table 2). Among a total 5 patients who had contrast CT (with or without PET), the cardiac lesions were broad-based/sessile in 3 patients and pedunculated in 2 patients. For 2 patients (patient #4 and #5), the right atrial lesion was initially overlooked as a contrast mixing artifact.

Upon further evaluation, only 2 patients (patient #2, 5) were symptomatic at the time of cardiac mMCC diagnosis. The other 7 patients initially denied cardiac symptoms, but 2 of 7 (patient #4, 8) eventually developed cardiac complications such as pericardial effusion and atrial fibrillation due to progression of cardiac MCC, and both patients died from heart failure (Table 3).

Nearly all initial cardiac lesions (8 of 9 cases) were observed in the right atrium and/or adjacent pericardium; only 1 patient presented with a cardiac lesion in the left atrium. Cardiac metastasis usually occurred as a later distant metastatic event. The median time between initial MCC diagnosis and first distant metastasis was 274 days [range 0–1494], while the interval between initial diagnosis and cardiac mMCC was 925 days [range 0–1494]. Only 1 patient had cardiac mMCC at the time of initial MCC diagnosis. The remaining 8 patients developed cardiac mMCC after 1 or more events of metastases (regional; *n* = 2, distant; *n* = 6). Of these 8, 5 patients had received systemic therapy prior to developing cardiac mMCC (Table 2).

Median follow-up after diagnosis of cardiac mMCC until death or last follow-up date was 325 days (range 85–2596). A detailed clinical vignette for a representative patient is presented in Figure 1.

### 3.3. Treatment for Cardiac mMCC

Patients were heterogeneously treated for cardiac mMCC at their discretion of their oncologist (Table 3). Five of 9 patients (patient #2, 3, 4, 5, and 7) received both radiotherapy to the cardiac lesion and immunotherapy such as programmed death-1 (PD-1) pathway blocking agents and/or anti-cytotoxic T lymphocyte antigen-4 (CTLA-4) agents. The other 4 patients received immunotherapy alone (patient #8), somatostatin analog alone (patient #6; immunotherapy was not available at that time), and chemotherapy (patient #1 developed cardiac mMCC while on immunotherapy; patient #9 initiated chemotherapy when immunotherapy was not yet available but later switched to immunotherapy). Figure 2 describes each patient’s disease trajectory, the timing of cardiac metastasis, and treatment. No patients developed therapy-related cardiac complications such as radiation- or immunotherapy-induced pericarditis.

### 3.4. Prognosis of Patients with Cardiac mMCC

Overall, 6 of 7 patients treated with immunotherapy achieved objective response in the cardiac lesion, either complete response (CR; *n* = 5) or partial response (PR; *n* = 1). Among these 6 patients, 1 patient (patient #8) treated with immunotherapy alone developed a local recurrence at the site of the initial cardiac lesion about 2 years after CR. The other 5 patients (patient #2, 3, 5, 7, 9) treated with a combination of immunotherapy and cardiac radiotherapy of 20–25 Gray (Gy) had durable local disease control, and only one of them (patient #5) developed local recurrence in the treated cardiac lesion. The one patient (patient #4) who did not respond to immunotherapy received 8Gy cardiac RT concurrently and rapidly declined. The remaining 2 patients (patient #1, 6) treated without immunotherapy/cardiac RT also developed local disease progression. The median duration of local control of the initial cardiac lesion was 679 days. Local control rates were 56% at 1 year and 44% at 3 years.

At the data cut-off date, 2 of 9 patients were still alive (1 had ongoing CR, 1 relapsed with a new non-cardiac metastasis 4 years after immunotherapy discontinuation in the setting of CR), and 7 have died. OS was 44% and 15% at 1 year and 3 years, respectively, following cardiac metastasis (Figure 3). A total of 296 non-cardiac mMCC patients were matched to our cardiac mMCC cohort. Each of the 9 cardiac mMCC patients had at least one matching non-cardiac mMCC patient, with 3—109 matches per cardiac mMCC patient [median: 13]. In the matched non-cardiac mMCC cohort, the median OS was 257 days, with OS at 1 year and 3 years of 45% and 36%, respectively. As shown in Figure 3, there was a substantial overlap between the 95% CIs for OS among the cardiac mMCC and matched non-cardiac mMCC patients. The difference between survival in the two cohorts was not statistically significant (*p* = 0.96 by stratified log-rank test).

### 3.5. Review of Published Literature

There were 14 unique patients with cardiac mMCC reported in the published case reports between 1990 to 2019 (Table 4) [14,15,16,17,18,19,20,21,22,23,24,25,26,27]. The median age at diagnosis was 67.5 years (range 23–82 years) with a male predominance (male 9, female 5). The most common site of cardiac metastasis was the right side of the heart (*n* = 11, 79%; right atrium *n* = 8, 57% and/or right ventricle *n* = 4, 29%). The most common primary site was the extremity (*n* = 6; 43%), followed by head/neck (*n* = 4; 29%). Cardiac metastasis was the first distant metastasis event for 8 of 14 patients (57%) and the second event for 6 (43%). Cardiac mMCC was detected simultaneously with other distant metastasis sites (lung, pancreas, stomach) in 4 patients.

Of the 14 previously described cases, 9 patients were treated for cardiac mMCC with systemic therapy (7 received chemotherapy, 2 immunotherapy) and/or radiotherapy. Immunotherapy was not available when these 7 patients were initially treated with chemotherapy. In the remaining 5 patients, 2 had radiotherapy alone, 1 underwent surgery, and 2 patients received no treatment—one due to a combination of age, comorbidities, and economic status, and the other because the patient died within days of cardiac mMCC diagnosis.

Approximate survival data, including follow-up time and survival status, were reported for 10 of 14 patients. Six of 10 patients died within 1 year of cardiac mMCC diagnosis, one patient died approximately 4.5 years after cardiac mMCC diagnosis, and three patients survived at least 8–12 months after cardiac mMCC diagnosis. OS at 1 year for the previously reported patients was estimated as 34%.

## 4. Discussion

Though a rare event in a rare disease, our case–control study and comparison with previously reported patients highlight several important patterns in cardiac metastases from MCC. Management of mMCC has advanced over the last 5–10 years. Immunotherapy has improved progression-free survival (PFS) and OS compared to conventional chemotherapy, which was associated with short durability (median PFS 90 days) and life expectancy (median OS 9.5 months) [1,37,38,39,40,41,42]. Avelumab, an anti-programmed death-ligand 1 (PD-L1) inhibitor, was the first FDA-approved drug for MCC in March 2017, and pembrolizumab, an anti-PD-1 inhibitor, was approved in December 2018. All patients with cardiac mMCC within our cohort were diagnosed between 2014–2021, and all but 3 were diagnosed in 2018 or later (i.e., in the era of immunotherapy) despite an eligibility period of 2003–2021. Moreover, cardiac involvement by MCC was a late, distant event. With improved survival associated with immunotherapy, late spread of MCC, including cardiac metastases, may be increasingly observed as patients survive longer. Our study is, therefore, timely to increase awareness of this pattern of relapse and to provide early therapeutic data. Despite the atypical location, cardiac involvement does not appear to be associated with worse survival compared to MCC with non-cardiac distant disease.

The majority of patients in our cohort were asymptomatic, and all were diagnosed incidentally by imaging, suggesting that cardiac metastases from MCC are likely underdiagnosed. This may be true of cardiac metastases from cancer in general. In a large autopsy series of over 18,751 patients, metastatic disease involving the heart was identified in 622 (3.3%) patients, with most frequent primary cancers including mesothelioma, melanoma, lung carcinoma, and breast carcinoma [43]. The pericardium was the most frequent site of cardiac metastasis (69.4%), followed by the epicardium (34.2%), myocardium (31.8%), and endocardium (5%) [43]. Metastasis to the heart has been reported through three mechanisms; (1) direct invasion, (2) lymphatic spread or (3) hematogenic spread [43]. Pericardial involvement is thought to be the result of either direct invasion by an intrathoracic or mediastinal tumor or retrograde lymphatic invasion through tracheal or broncho mediastinal lymphatic channels, while endocardial metastases stem from hematogenic spread [43]. Contrary to what was observed in the autopsy series, all patients in our cohort had endocardium lesions. Both within our series and previously reported cases, the predominance of endocardial lesions involved in the right side of the heart supports the hypothesis that cardiac mMCC disseminates via hematogenous spread [14,15,16,17,18,19,20,21,22,23,24,25,26,27]. The predilection of the right atrium/ventricle may be a characteristic feature of cardiac metastasis in MCC. It is also of interest that extremity was the most common primary site among both our cohort and the previously reported cases [14,15,16,17,18,19,20,21,22,23,24,25,26,27].

In the absence of a standard approach, we advocate for multidisciplinary care with the goal to palliate symptoms if present and prevent or delay symptom recurrence. Beyond utilization of immunotherapy, which had evidence of efficacy within our cohort (6 out of 7 with CR or PR), local therapy with cardiac radiotherapy can also be incorporated, given the radio responsiveness of MCC and its potential synergism with immunotherapy [1,44,45,46]. While single fraction 8Gy cardiac RT was insufficient due to lack of local disease control in the cardiac lesion, moderate radiation doses of 20–25 Gy (given over 5–8 fractions) were associated with clinically meaningful, durable disease control (CR or PR). Almost all patients who received 20–25Gy did not have local disease progression, including those who developed cardiac MCC while on immunotherapy.

While our study is limited by small numbers, adding radiotherapy to immunotherapy was not associated with increased cardiotoxicity. No patients in this cohort or in the reported literature experienced cardiac complications due to radiation or immunotherapy. There may be the potential for “treatment inertia” with immune checkpoint inhibitors, due to potential risk for immunotherapy-associated myocarditis, but none of the patients in our series experienced myocarditis. Additionally, there may be hesitancy to pursue cardiac-directed radiation due to risk for acute cardiotoxicity including inflammatory related changes such as myocarditis or pericarditis. In our series, no patients experienced any acute adverse events (AEs), arguing that with careful radiation therapy planning, focused cardiac-directed radiation therapy may be well-tolerated. Instead, cardiac complications (including pericardial effusion and atrial fibrillation) were encountered at the time of cardiac mMCC diagnosis or disease progression.

Radiologic imaging plays a critical role in diagnosis and follow-up of cardiac mMCC [47,48]. All patients in our series were incidentally noted to have a cardiac mass by routine surveillance or staging imaging studies, such as FDG PET/CT, CT, and/or magnetic resonance imaging (MRI). Once cardiac involvement is suspected, further imaging workup can be pursued with transthoracic echocardiography (TTE), a readily available, noninvasive imaging technique [47,49]. On contrast-enhanced CT, intracardiac lesions can often be seen as a filling defect within the cardiac chambers [47]. Cardiac MRI can be utilized to pursue noninvasive tissue characterization, since cardiac tumors show low intensity on T1-weighted images and intermediate to high intensity on T2-weighted images [49,50]. Additionally, cardiac tumors have heterogeneous gadolinium enhancement on cardiac MRI, which is an important feature in differentiating the mass from thrombus [50]. Furthermore, resting first-pass perfusion can assess for vascularity as a clue to presence of tumor [47,49]. These findings were observed in all our patients who underwent a cardiac MRI study. Though an endomyocardial biopsy is needed for definitive confirmation, this invasive procedure can be associated with complications and is also subject to sampling error and false negatives [51]. Therefore, a cardiac biopsy is often deferred, and the diagnosis of cardiac mMCC is most often made by imaging, especially when the patient already has confirmation of distant MCC elsewhere.

An important radiological differential diagnosis for cardiac mass is benign thrombus, which may be difficult to differentiate on echocardiogram or CT. Cardiac MRI or FDG PET/CT scan is useful for differentiation of thrombus from tumor [47], given that, in contrast to a thrombus, cardiac MCC is associated with intralesional enhancement or high FDG uptake, respectively. An alternative imaging technique is a somatostatin-seeking nuclear medicine study, as the majority of MCC (85%) exhibits somatostatin receptors on the tumor cell surface, which may be useful to differentiate from other malignant etiologies [48,52].

Several limitations should be noted within this study, including first, the modest patient numbers of this rare entity. Despite this, to our knowledge, this represents the largest series of MCC cardiac metastases to date. Second, due to the retrospective nature of this study, management and workup of patients was heterogeneous. However, there are no established standard of care treatments for these patients, and the varied management allowed us to identify potentially efficacious treatment paradigms.

## 5. Conclusions

Cardiac mMCC generally occurs later in the disease course, and most commonly involves the right side of the heart, in particular the right atrium. Involvement of the heart was not associated with worse survival compared to other distant metastasis sites. We found that immunotherapy and moderate dose of cardiac radiotherapy (20–25 Gy) were associated with high rates of response and were well tolerated.

## Figures and Tables

**Figure 1 cancers-14-05914-f001:**
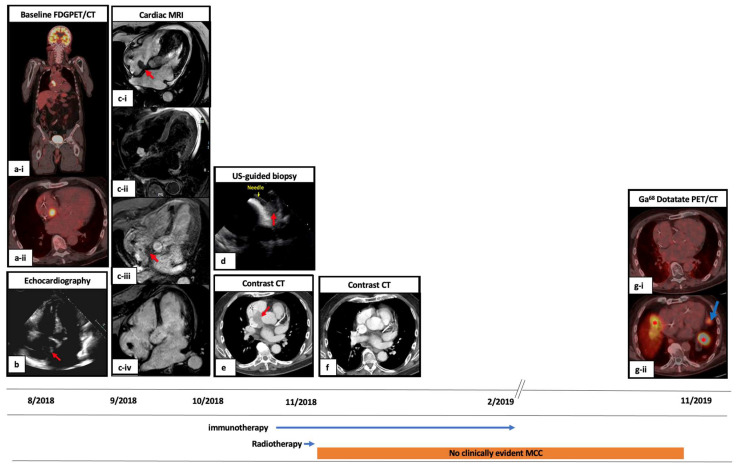
A case of patient with Merkel cell carcinoma of the buttock and oligo-metastasis to the heart. An 86-year-old immunocompetent man was initially diagnosed with a 10 cm Merkel cell carcinoma (MCC) of the left buttock. His baseline fluorodeoxyglucose/positron emission tomography/computed tomography (FDG PET/CT) for staging during the initial workup showed an incidental abnormal finding of a hypermetabolic mass in the right atrium (panel **a-i**) and the intense FDG avidity on PET/CT made a myxoma less likely as the cause (panel **a-ii**). The lesion was then evaluated by cardiac ultrasound (panel **b**). On subsequent cardiac magnetic resonance imaging (MRI), there was an irregular mass in the right atrium that was attached to the interatrial septum and extended superiorly to the distal superior vena cava (panel **c-i**). This mass demonstrated high signal intensity on the T2 black-blood sequence (panel **c-ii**), weak perfusion on perfusion MRI (panel **c-iii**), and late gadolinium enhancement on late postcontrast sequence (panel c-iv), which indicated that this did not represent a thrombus. The cardiac mass was biopsied under intracardiac echocardiography (panel **d**) with pathology confirming metastatic MCC. The patient started first-line systemic therapy with avelumab followed by palliative radiation therapy to the primary MCC of the left buttock with 30 Gray (Gy) in 10 fractions and to the right atrium mass with 20 Gy in 8 fractions. Compared to contrast CT performed before RT (panel **e**), a restaging CT scan performed 1 month after RT completion showed complete resolution of the right cardiac MCC mass (panel **f**). His primary MCC disease on the buttock also had a complete response. He discontinued avelumab due to grade 3 immune-related pneumonitis 8 months after, without clinically apparent evidence of MCC. Unfortunately, approximately 6 months after avelumab discontinuation, he was found to have a new enhancing left pericardial lesion measuring 1.9 × 1.2 cm in size on contrast CT outside the radiation field. On a Gallium-68 (Ga^68^) Dotatate PET/CT, there was no uptake in the initially treated right atrium (panel **g-i**), but there was a new intense radiotracer uptake in the left pericardial lesion (blue arrow. Red * showed physiological uptakes) (panel **g-ii**). He had a single 8 Gy fraction of radiation therapy to the new left pericardial mass, but the lesion progressed. Monthly somatostatin analog therapy was initiated but discontinued after 7 doses due to disease progression. The patient then switched to combination therapy with ipilimumab and nivolumab, but he passed away from non-MCC causes 6 months after initiating the combination immunotherapy. The red arrow shows the cardiac mMCC in the right atrium.

**Figure 2 cancers-14-05914-f002:**
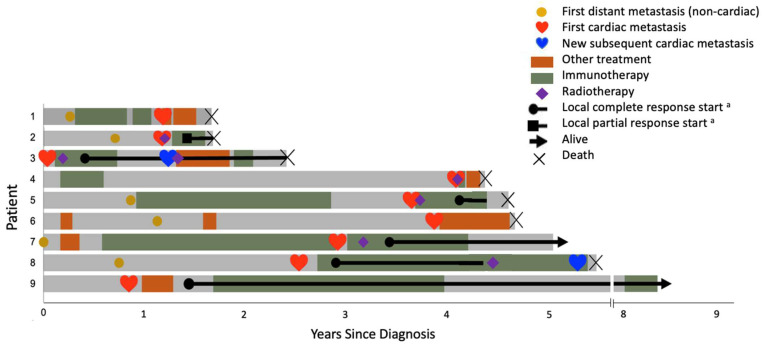
Swimmer plot of the clinical course of each patient. Swimmer plots with each lane representing an individual patient’s disease trajectory, the timing of cardiac metastasis, treatment, and kinetics of response and disease progression. The heavy black lines represent responses to cardiac metastasis-directed treatment. Patient #3 developed the new subsequent cardiac metastasis outside the radiation field in the left side of the heart and did not recur in the initial treated cardiac lesion. The clinical course of this patient is described in Figure 1. Patient #8 received immunotherapy without radiation. Although Patient #8 achieved complete response including the initial lesion in the right side of the heart, the patient had a local cardiac recurrence about 2 years later and eventually developed a new metastasis in the left side of the heart. (a) Local response at the treated 1st cardiac mMCC.

**Figure 3 cancers-14-05914-f003:**
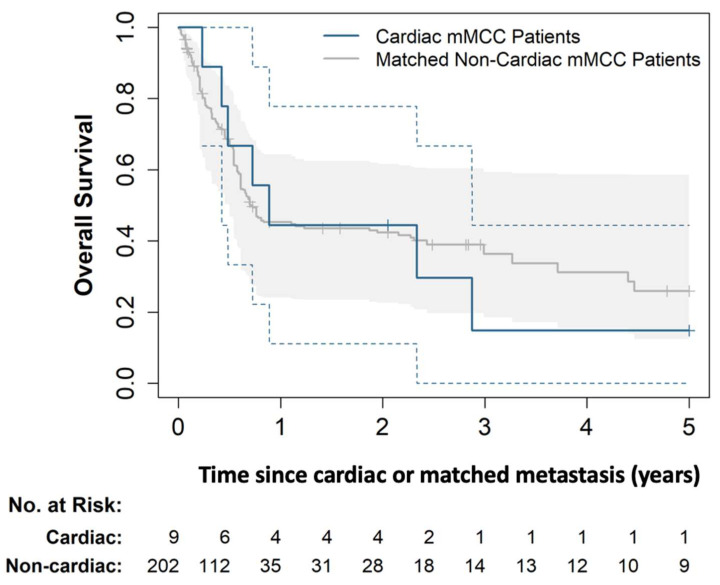
Overall survival among cardiac mMCC patients and non-cardiac mMCC patients matched by number of prior metastases. Overall survival (OS) of the cardiac mMCC patients (blue line) and matched non-cardiac mMCC patients (gray line). The OS time was calculated starting from the date of cardiac mMCC diagnosis for the cardiac mMCC patients and starting from the date of the matching non-cardiac mMCC diagnosis for the non-cardiac mMCC patients. The matching non-cardiac mMCC diagnosis was the one with the same number of prior metastatic episodes as the matched cardiac mMCC patient at the time of the cardiac mMCC diagnosis (see the Appendix A for more detail on the matching). The 95% confidence intervals for the two survival curves are shown using the blue dashed lines and solid gray region, respectively. The tick marks on each curve indicate censoring times. The two curves were not statistically significantly different (*p* = 0.96 by the stratified log-rank test).

**Table 1 cancers-14-05914-t001:** Patient, pathologic, and treatment characteristics at initial Merkel cell carcinoma diagnosis.

Case	Age	Sex	Immune Suppression	Primary Site	Initial Stage	Initial Treatment	MCPyV Status ^a^
1	52	M	No	Unknown primary	pIIIA	Excision	Positive
2	69	M	Yes ^b^	Extremity	pI	WLE, SLNB, RT	Negative
3	86	M	No	Trunk	pIV	Avelumab, RT	Positive
4	64	F	No	Extremity	pIIIB	SLNB, RT, Nivolumab ^c^	Positive
5	61	M	No	Head/neck	pIIIA	WLE, SLNB, RT	Negative
6	69	M	No	Unknown primary	pIIIA	Parotidectomy, neck dissection, RT, chemotherapy ^d^	Negative
7	71	M	No	Extremity	pIV	Excision, SLNB, RT, Chemotherapy ^e^	Positive
8	76	F	No	Extremity	pIIIB	WLE, lymphadenectomy, RT	Positive
9	62	M	No	Head/neck	pIIIB	WLE, SLNB, RT	Positive

a: Virus status assessed by MCPyV T-Ag oncoprotein antibody serology assay or by tumor immunohistochemistry using anti-MCPyV T-Ag antibody (CM2B4). Patient #5 and #6 were negative for MCPyV by both serology assay and immunohistochemistry assessment. Patient #2 was seronegative, but immunohistochemistry was not performed, b: Treated with adalimumab for Crohn’s disease, c: Initiated adjuvant nivolumab due to unresectable, in-transit metastasis, d: Cisplatin/etoposide, e: Carboplatin/etoposide. Patients treated with chemotherapy (#6 and 7) were diagnosed with Merkel cell carcinoma (MCC) before immunotherapy became available. Chemoradiation was performed for advanced unresectable MCC (#6) and for metastatic MCC (#7). M: male, F: female, WLE: wide local excision, SLNB: sentinel lymph node biopsy, RT: radiotherapy, MCC: Merkel cell carcinoma, MCPyV: Merkel cell polyomavirus.

**Table 2 cancers-14-05914-t002:** Imaging details and timing of cardiac metastatic MCC.

Case	Site of Initial Cardiac Metastasis	Imaging Modality to Initially Detect Cardiac mMCC	Additional Imaging Workup of Cardiac mMCC	Days from Initial MCC Diagnosis to First Distant Metastasis	Days from Initial MCC Diagnosis to Cardiac mMCC	Site of Non-Cardiac Metastases at the Time of Cardiac Metastasis Diagnosis	Systemic Treatment Prior to Cardiac Metastasis
1	Left atrium	FDG PET/CT	None	96	432	Right groin LN, right iliacus muscle lesion, right testicle	Pembrolizumab, nivolumab/ ipilimumab, pazopanib
2	Right atrium	FDG PET/CT	None	254	428	Right arm soft tissue, retroperitoneum, right supraclavicular, cervical, and mediastinal LNs, right humerus	None
3	Right atrium	FDG PET/CT	TTE, contrast CT	0	0	None	None ^a^
4	Right atrium, interatrial septum	Contrast CT	None	1494	1494	Left thigh soft tissue, left inguinal LN	Nivolumab ^b^
5	Right atrium extending into right ventricle	Contrast CT	MRI, TTE	316	1333	Bones, pelvic/peritoneal soft tissue mass	Nivolumab/ ipilimumab
6	Right atrium	FDG PET/CT	Octreoscan SPECT/CT	412	1415	Left maxillary sinus, left cervical/ supraclavicular LN, right adrenal gland	Cisplatin/etoposide ^c^
7	Right atrium	FDG PET/CT	MRI	0	1065	None	Carboplatin/ Etoposide ^d^, avelumab
8	Right atrium	FDG PET/CT	Contrast CT	274	925	Right external iliac, paraaortic, and retrocrural LNs	None ^e^
9	Right atrium	Contrast CT	FDG PET/CT, echo- cardiogram	309	309	Pancreas, left upper quadrant abdominal mass	None

a: Patient presented with stage IV MCC of the buttock involving a single distant metastasis to the heart. b: Nivolumab was initiated for unresectable in-transit metastasis. c, d: These patients received chemotherapy prior to cardiac metastasis since immunotherapy was not available at that time. Patient #7 switched to avelumab later when avelumab was approved for MCC. e: Non-cardiac metastatic lesions were initially treated with surgical excision or radiotherapy. No systemic treatment was initiated before the cardiac mMCC. The date of distant metastasis, including cardiac metastasis after the initial treatment, is defined when abnormal findings were noted on restaging imaging or when patients had a biopsy if performed. CT: computed tomography, FDG PET/CT: fluorodeoxyglucose positron emission tomography/computed tomography, LN: lymph node, mMCC: metastatic Merkel cell carcinoma, MRI: magnetic resonance imaging, SPECT: single-photon emission computerized tomography, TTE: transthoracic echocardiogram.

**Table 3 cancers-14-05914-t003:** Summary of treatment for cardiac mMCC and outcomes.

Case	Treatment for Cardiac and Other mMCC	Year Cardiac mMCC Diagnosed	Cardiac Complications Due to Cardiac Recurrence or Disease Progression	Best Objective Response for 1st Cardiac mMCC	Days after Cardiac mMCC Diagnosis to Local Recurrence or Progression in the Treated 1st Cardiac mMCC	Days after Cardiac mMCC Diagnosis to First Recurrence or Progression at Any Site	Survival Status and Cause of Death	Overall Survival ^a^ (Days after Cardiac mMCC Diagnosis)
1	Carboplatin/ Etoposide ^b^	2021	None	PD	29	29	Deceased, MCC	177
2	RT 20 Gy in 5 fractions to heart, followed by avelumab infusion	2018	Shortness of breath due to pericardial effusion, tumor thrombus in coronary sinus, atrial fibrillation	PR	N/A, No initial cardiac mMCC progression prior to death	91	Deceased, MCC	155
3	Avelumab for 8 months ^c^, RT 20 Gy in 8 fractions to heart	2018	None	CR	N/A, no recurrence in the initial cardiac mMCC lesion prior to death	451	Deceased, non-MCC	853
4	RT 8 Gy in 1 fraction to heart, Cavrotolimod/pembrolizumab	2021	Pericardial effusion, Tachycardia- bradycardia syndrome	PD	47	47	Deceased, MCC	85
5	Ipilimumab for 2 months and Nivolumab ongoing until death. RT 20 Gy in 5 fractions to heart	2019	Pericardial effusion	CR	269	89	Deceased, MCC	325
6	Sandostatin ^d^	2014	None	PD	112	112	Deceased, MCC	265
7	Nivolumab/ ipilimumab, followed by MR-guided adaptive RT 25 Gy in 5 fractions	2019	None	CR	N/A, ongoing CR	N/A, ongoing CR	Alive	749
8	Pembrolizumab	2015	Heart failure	CR	679 ^e^	679	Deceased, MCC	1050
9	Carboplatin/ etoposide for 4 months with PD ^f^. Switched to avelumab and was on avelumab for 2 years and 4 months	2014	None	CR	N/A, no recurrence in the initial cardiac mMCC lesion	2575 ^g^	Alive	2596

a: Overall survival is defined as “days from the date of cardiac mMCC diagnosis to the date of death”. Patients who were alive at the time of data cut-off were censored at the date last known to be alive. b: Prior to developing cardiac mMCC, the patient was treated with pembrolizumab with progression disease, then switched to a combination therapy of nivolumab and ipilimumab with progression disease. Thus, chemotherapy was initiated. c: Developed grade 3 immune-mediated pneumonitis 8 months into avelumab treatment. d: Immunotherapy was not available at that time. e: Achieved complete response to pembrolizumab monotherapy without radiotherapy, including the initial lesion in the right side of the heart. However, the patient had a local cardiac recurrence at the site of the initial right-sided heart lesion about 2 years later and eventually developed a new metastasis in the left side of the heart. f: Immunotherapy was not available, and the patient first started systemic therapy with carboplatin and etoposide for metastatic MCC lesions in the heart and pancreas with a progressive disease of the pancreatic lesion. g: Developed biopsy-proven metastatic MCC to the perinephric area 4 years after avelumab discontinuation in the setting of no clinically evident disease. CR: complete response, Gy: Gray, MCC: Merkel cell carcinoma, mMCC: metastatic Merkel cell carcinoma, MR: magnetic resonance, OS: overall survival, PD: progressive disease, PFS: progression-free survival, PR: partial response, RT: radiation therapy, N/A: not applicable.

**Table 4 cancers-14-05914-t004:** Summary of 14 previously published cases.

Literature	Age	Sex	Comorbidity	Primary Site	Nodal or Distant Involvement at Initial MCC Diagnosis	Duration from Initial Diagnosis to Cardiac Metastases	Site of Cardiac Metastasis	Site of Other Metastases Diagnosed at the Time of Cardiac Metastasis	Treatment for Cardiac mMCC	Cardiac Complications Due to Cardiac mMCC
Chao et al., 1990 [14]	23	F	Pregnancy	Trunk	No	1 year, 2 months	R ventricle	None	RT, chemotherapy	Grade IV systemic murmur
Page et al., 2001 [24]	72	F	Unknown	Head/ neck	No	~1 year	R and L ventricles	Lungs	Chemotherapy	Unknown
Jongbloed MRM et al., 2004 [20]	63	F	Unknown	Extremity	Yes	~3 years	R atrium, R ventricle	None	None. Died a few days after diagnosis of cardiac mMCC	Cardiac tamponade
Conley M et al., 2006 [15]	66	M	Unknown	Extremity	No	~5 years	R atrium	None	Hematopoietic cell transplant, melphalan	Atrial fibrillation, cardiac tamponade, complete heart block
Fiorillo J, 2008 [17]	76	M	Unknown	Extremity	No	~8 years	R and L atria	None	Bortezomib/ melphalan	Pericardial effusion
Keeling A et al., 2010 [22]	63	M	Unknown	Testicle	Yes	Unknown	R atrium	None	Resection	Unknown
Fong L et al., 2012 [18]	80	M	Unknown	Extremity	Yes	Unknown	R atrium	None	RT	Unknown
Yamana N et al., 2013 [27]	54	F	Unknown	Head/ neck	No	~3.5 years	R atrium, interatrial septum	None	Cisplatin/etoposide with PD, RT 43 Gy in 19 fractions	Dyspnea, epigastralgia
Wang L et al., 2014 [26]	76	M	Unknown	Head/ neck	Not described, but treated with radical neck dissection	~2 years	Coronary sinus	LN adjacent to the pancreas	Carboplatin/ etoposide	Cardiac lesion encased a left ventricular pacing lead, which led to ventricular tachycardia
Suttie et al., 2014 [25]	79	M	Unknown	No known primary	Yes	11 months	R atrium	None	RT	Dyspnea on exertion
Mantripragada & Birnbaum, 2015 [23]	40	M	Prior chemotherapy for another malignancy	Head/ neck	Yes	~1 year	R Ventricle, posterior intra-atrial septum	Pancreas, pericardial lymph node	Nivolumab	None
Di Loreto M et al., 2017 [16]	59	M	Non-Hodgkin’s lymphoma	Extremity	No	~2 years	R atrium, pericardial space	None	Cisplatin/ etoposide	Cardiac tamponade
Ha J et al., 2018 [19]	82	M	None	Trunk	Not described	~3 years	L atrium, interatrial septum	Stomach	None. Died on palliative care	Mild pericardial effusion
Kazemi N et al., 2019 [21]	73	F	Non-Hodgkin’s lymphoma	Extremity	Yes	~1.5 years	interatrial septum	None	Avelumab, RT 40 Gy in 5 fractions	Second- degree AV block (Mobitz type II) requiring pacemaker placement

AV block: Atrioventricular block, F: female, Gy: gray, L: left, M: male, MCC, Merkel cell carcinoma, mMCC: metastatic Merkel cell carcinoma, R: right, RT: radiotherapy.

## Data Availability

Data are contained within the article and Appendix A.

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
