# Peer review of "Management and Prognosis of Cardiac Metastatic Merkel Cell Carcinoma: A Case–Control Study and Literature Review"

_cancers, 2022, doi:10.3390/cancers14235914_

Round 1

Reviewer 1 Report

The present work is a Case-Control Study entitled “Management and Prognosis of Cardiac Metastatic Merkel Cell 2 Carcinoma: A Case-Control Study and Literature Review”. The work evaluated a prospective registry of 582 patients with metastatic MCC (mMCC), and authors identified 9 patients (1.5%) with cardiac involvement. These 9 cases were described in detail. A review of previously published cases of cardiac metastatic MCCs has also been conducted.

The paper is interesting and well written. However, several points sohuld be addressed.  I have few comments
The authors may consider the following points:

1. Study limitations should be included.
2. The introductive section should be improved with more details of the MCC metastatic potential in different anatomic sites, including the heart.
3. Despite the fact that about 20% of MCC cases are prone to metastasize, as stated in the abstract, the authors identified about 30%  of patients carry distant metastases. This point should be clarified.  
4. The results of the literature review are not discussed. The high number (n=9) of heart metastasis MCC cases in the study cohort from only one US Cancer Research Center, paralleled with a few anecdotal cases (only n=9) reported in the literature, is a very and important point that should be underlined in the discussion. A multicentric investigation should be conducted in the future for evaluating the impact of heart metastatic MCC.

Moreover, please see below several suggestions for improving the work.
1) Lines 48-50 About 80% of MCCs are MCPyV-drive, globally. The sentence should be rephrased. Moreover, line 46, concerning the MCC onset risk related to conditions of immune suppression, this supporting reference should be included. DOI: 10.1158/1078-0432.CCR-16-2899
2) Line 50 and UV-induced point mutations
3) If the registry came from the Fred Hutchinson Cancer Research Center (FHCRC) I suggest including its official website and/or registry code. City and state of the research center should also be detailed.
4) Statistical analysis section is completely lacking in supporting references.
5) If available, it would be interesting including in table 1 some information on the jobs and/or lifestyle of the 9 MCC affected patients. MCC it is well known to arise in chronically sun-exposed individuals (https://www.cancer.org/cancer/merkel-cell-skin-cancer/causes-risks-prevention/risk-factors.html), e.g, farmers.
6) Supplementary material should have been submitted in a separate file

Author Response

We thank the reviewer for the comments. Please see the attachment. 

Reviewer 2 Report

The authors of this manuscript investigate the occurrence, prognosis, as well as potential viable treatment strategies of cardiac metastatic Merkel Cell Carcinoma (mMCC) patients. Through a cohort study of 582 mMCC patients, 9 patients were identified to have cardiac metastases. It was determined that cardiac metastases had a predilection for the right side of the heart (right atrium/ventricle) and often occurred late after initial MCC diagnosis. It was discovered that patients responded well to a combination of immunotherapy and cardiac radiotherapy that could prevent local disease progression, suggesting that this could be a potential treatment method for cardiac mMCC patients. Some of these results were corroborated through a primary literature search, which identified 14 patients with cardiac mMCC. Though these results are preliminary and based off a small sample size, this manuscript highlights some key features of cardiac mMCC, which has not been well reported previously. This report is of importance as MCC patients receiving immunotherapy are expected to have a longer lifespan and therefore could have a higher likelihood of getting cardiac metastasis due to its occurrence later on during MCC progression.    - Since the Figure 1 shows a detailed clinical vignette for a representative patient, it would be nice to show the pathologic data confirming mMCC (as authors mentioned in the figure legend), if available.   - There were some slight formatting issues that could be fixed so that printed copies of the paper can be more easily read: 
  • Please check lines 102-103 (~"heart,"" metastasis," or "metastatic.")
  • Some letters in the Tables are hard to read because of automatic text wrap. The font size could be adjusted or the tables could be added as a figure file.
  • The header for Table 2 (located on page 4) is cut off from the rest of Table 2 (located on page 5). Moving the Table 2 header to page 5 with the rest of the Table would make it easier to comprehend the data presented in the table.
  • Result Section Header 3.3. Treatment for cardiac mMCC (page 8) could be moved to the next page (page 9) so it is included with the Result section body.

Author Response

We thank the reviewer for these comments. Please see the attachment.

Round 2

Reviewer 1 Report

Now the article has been improved and may be considered for publication.